# Evaluation of Food-Grade Additives on the Viability of Ten *Shigella flexneri* Phages in Food to Improve Safety in Agricultural Products

**DOI:** 10.3390/v17040474

**Published:** 2025-03-26

**Authors:** David Tomat, Cecilia Casabonne, Virginia Aquili, Andrea Quiberoni

**Affiliations:** 1Área de Bacteriología, Facultad de Ciencias Bioquímicas y Farmacéuticas, Universidad Nacional de Rosario, Rosario S2002LRK, Argentina; ccasabonne@fbioyf.unr.edu.ar (C.C.); viraquili@hotmail.com (V.A.); 2Instituto de Lactología Industrial (UNL—CONICET), Facultad de Ingeniería Química, Santa Fe 3000, Argentina; aquibe@fiq.unl.edu.ar

**Keywords:** *Shigella flexneri*, bacteriophage, food additives, biocides, viability

## Abstract

Bacteriophages can be used as biocontrol agents in agriculture to improve food safety, provided they can remain viable in food environments. The viability of ten *Shigella* phages (AShi, Shi3, Shi22, Shi30, Shi33, Shi34, Shi40, Shi88, Shi93, and Shi113) was evaluated against different additives and biocides used daily in food applications. In addition, the influence of additives on phage viability in a food matrix was investigated. Treatments with lactic and citric acid were the most effective to inactivate phages. In addition, the acetic acid was the most phage-friendly treatment evaluated. Preservatives such as acetate, lactate, benzoate, sorbate, and propionate proved to be highly compatible with all the phages tested. Regarding the influence of the food matrix on phage viability, an equal or higher viability was found for most phages tested when compared with the corresponding organic acid. Finally, when phages were exposed to sodium hypochlorite, ethanol, quaternary ammonium chloride (QAC), and H_2_O_2_, most of them were sensitive to long incubations and high concentrations. However, when biocide concentrations employed are low, 10^3^–10^4^ PFU mL^−1^ phage particles remains viable. Thus, the phages evaluated could be used in combination with additives and biocides as a biocontrol tool against the foodborne pathogen *S*. *flexneri* in agricultural products.

## 1. Introduction

*Shigella* spp. are foodborne and waterborne pathogens [1,2] that have been isolated from different foods [3] such as vegetables, chicken, produce, and dairy products [4,5]. Developing countries have a high incidence of Shigellosis [6] that specially affects children under five-years-of-age [7]. Specifically, *Shigella flexneri* is the most prevalent pathogen responsible for Shigellosis in Argentina [8]. This bacterium is accountable for food poisoning outbreaks worldwide [9] and is an important cause of infant mortality in our country [10].

The treatment of food with additives such as organic acids is an effective approach to reduce populations of pathogens [11]. However, *Shigella* can survive for significant periods of time under conditions such as a high concentration of weak organic acid [12,13]. These acids are the most employed food-grade additives in Argentina; thus, phages must survive in their presence in order to be an efficient tool to control foodborne pathogens. Also, biocides like hypochlorite and ethanol are commonly used in the food industry to decontaminate and sanitize food related surfaces such as equipment and tools. Bacteriophages are biocontrol tools that may be used as an economic and more natural technology to improve food safety. Previous studies conducted with phages active against foodborne pathogens such as *Shigella* [14,15] and *E. coli* [16] demonstrate their potential for use in food safety. Unlike additives, phages are very specific agents that attack only the targeted bacteria without affecting the organoleptic properties of foods [16]. Therefore, a technological characterization of phages active against *Shigella flexneri* in different food environments to fight foodborne pathogens may allow us to select phages for more effective biocontrol treatments.

In our previous studies, ten phages lytic against *S*. *flexneri* strains were isolated and characterized by their host range, challenge and adsorption assays, and stability tests under various stress conditions [17,18]. Thus, the aim of the present work is to evaluate the influence of additives present in food and biocides present in food-related environments on phage viability. Although *Shigella* phage studies have been conducted on various foods [14,19,20], those focused on evaluating the viability of *S. flexneri* phages challenged with a combination of additives or everyday biocides have not been documented.

## 2. Materials and Methods

### 2.1. Bacterial Strains and Phages

The *Shigella flexneri* strain ATCC 12022 (serotype 2b) was used to propagate (phage stocks, Tris-magnesium gelatin; Tris-HCl 0.05 M (pH 7.5); MgSO_4_ 10 mM; gelatin 0.01% *w*/*v*, TMG, 4 °C) and to enumerate (plaque forming unit per milliliter; PFU mL^−1^) phages in viability studies. Bacterial stocks (Triptein soy broth, TSB, 15% *v*/*v* glycerol, −80 °C) were reactivated in trypticase soy broth at 37 °C overnight for viability assays with phages. Ten (10) phages (AShi, Shi3, Shi22, Shi30, Shi33, Shi34, Shi40, Shi88, Shi93, Shi113) previously isolated from 114 stool samples [17] were used to evaluate their viability against several concentrations of various biocides and preservatives.

### 2.2. Phage Viability Against Food Additives and Biocides

The viability of ten *Shigella* phages was evaluated against food-grade additives and different biocides. Phage suspensions (TMG buffer with 10^5^–10^6^ PFU mL^−1^) were challenged against weak organic acids, their salts, and biocides. Assays were conducted at 25 °C with a final volume of 1 mL. Table 1 details the treatments (additive or biocides, concentration; incubation time). The concentrations used are the maximum allowed (or lower) for each preservative in food [21]. After incubation with each additive or biocide, phages were enumerated by the double-layer plate titration method [22]. Plaques (Triptein soy agar, TSA, 1.5%) were incubated for 18 h at 37 °C and the PFU were counted to evaluate the viability of each phage. Results were expressed as PFU mL^−1^. Controls were performed in sterile distilled water without additives and biocides.

### 2.3. Phage Viability Against Food Additives in Food

To evaluate the phage viability of food preservatives in a food matrix, pieces (1 cm^2^; pH 5.7) of beef were cut aseptically, placed in Petri dishes, and pre-equilibrated to 25 °C. Next, 20 μL of each phage (~10^5^–10^6^ PFU mL^−1^) and 20 μL of each organic acid (4%) (acetic, lactic and citric) were pipetted onto the surface of the meat sample and allowed to dry (10 min at 25 °C) after the addition of each volume. To set the 100% of phage viability, samples with phages were also inoculated with 20 μL of TMG buffer instead of the organic acids. Controls and treatments were incubated at 25 °C. After each incubation time (30 min, 60 min), meat pieces were transferred to a sterile bag, 1 mL of TMG buffer was added, and samples processed for 2 min in a Stomacher (Seward, London, UK). Then, all the stomacher fluid (1 mL) was transferred to a sterile Eppendorf tube, centrifuged (3000 rpm, 10 min), and 0.1 mL of the supernatant was plated for phage enumeration as previously described [14].

### 2.4. Statistical Analysis

The mean values were calculated from three different experiments, then treatments (two technical replicates: mean of 6 data points) and controls (two technical replicates: mean of 6 data points) were compared using Student’s *t* test at *p* < 0.05.

## 3. Results

### 3.1. Phage Viability Against Food Additives

The effect of acetic acid in phage viability is shown in Figure 1. Three phages, AShi, Shi3, and Shi22, showed a high resistance (>10^2^ PFU mL^−1^) to this preservative since their viabilities were only slightly affected at the higher concentration (4%) tested after long incubation (30 and 60 min) times. AShi was the only phage that was detected (70 PFU mL^−1^) after 24 h of incubation at the lower concentration (2%) assayed. Regarding the other seven phages, Shi30, Shi33, Shi34, Shi40, Shi88, Shi93, and Shi113, a significant decrease in PFU mL^−1^ was observed at a low concentration (2%) of acetic acid after 30 and 60 min of incubation, although a significant number of phages particles (10^2^–10^3^ PFU mL^−1^) remained viable. In addition, at a high concentration of acetic acid (4%), phages (10^3^–10^4^ PFU mL^−1^) endured up to 15 min of incubation and a complete inactivation occurred after longer incubation times. No viability was detected when these seven phages were evaluated after a 24 h-incubation period.

Figure 2 shows the viability of phages evaluated against two concentrations (2 and 4%) of lactic acid. AShi and Shi3 proved to be the most resistant phages to this particular weak organic acid. Although a significant decrease was observed for these two phages at 2% of lactic acid, 10^2^ (PFU mL^−1^; Shi3) and 10^4^ (PFU mL^−1^; AShi) particles were detected at the end of the experiments (60 min). Moreover, at a high concentration of lactic acid (4%), the complete inactivation was produced only after 30 min of incubation. The remaining eight phages (Shi22, Shi30, Shi33, Shi34, Shi40, Shi88, Shi93, Shi113) were found to be more sensitive than AShi and Shi3. Namely, phage particles of Shi33, Shi88, and Shi93 were detected after a 5 min incubation, and then a complete inactivation was observed at both concentrations assayed, whereas for phages Shi22, Shi30, Shi34, Shi40, and Shi113 particles could be detected after a longer incubation time (15 min) at only the low concentration (2%) evaluated.

Regarding viability against citric acid, all the phages evaluated showed a moderate resistance (10^1^–10^2^ PFU mL^−1^) at 2% of this preservative since 10^2^ PFU mL^−1^ or more particles remained viable at up to 30 min of incubation (Figure 3). At 4%, a significant number of particles were detected after 30 min (AShi and Shi3), 15 min (Shi22), and 5 min (Shi30, Shi33, Shi34, Shi40, Shi88, Shi93, Shi113) of incubation. However, longer incubation times resulted in a complete inactivation of all the phages tested.

Taking into account the numbers of bars remaining after each phage treatment, the most effective acid to inactivate phages was the lactic followed in effectiveness by the citric. The acetic acid was the most phage-friendly treatment evaluated, i.e., a greater number of bars. Furthermore, preservatives such as acetate (sodium), lactate (sodium), citrate (sodium), benzoate (sodium), sorbate (potassium), and propionate (sodium) were also evaluated and no significant differences were observed in the viability of the ten phages tested compared to the control conditions.

### 3.2. Phage Viability Against Food Additives in Food

When phages were challenged against acetic acid in a food matrix such as meat, AShi, Shi3, and Shi22 were not affected since their viabilities were not significantly reduced at both incubation times evaluated. A similar behavior was only observed for the remaining phages after 30 min of incubation, where phage count remained in the same order of magnitude of the controls. On the other hand, when the viability of these phages (Shi30, Shi33, Shi34, Shi40, Shi88, Shi93, and Shi113) was assessed after 60 min, a significant reduction (2 log_10_ CFU mL^−1^) was observed for all of them; however, a significant number of particles (10^3^–10^4^ PFU mL^−1^) could still be detected.

Lactic acid produced a significant reduction in phage viabilities after 30 min of incubation, although half of the particles (10^3^–10^4^ PFU mL^−1^) remained viable for each phage evaluated. After a 60 min incubation, only AShi and Shi3 could be detected (10^2^–10^3^ PFU mL^−1^) since the other eight phages were completely inactivated.

Likewise, citric acid had a similar effect on phage viability as lactic acid. Namely, AShi and Shi3 were the most resistant phages to both organic acids. Treatments with citric acid at 4% resulted in the detection of ~10^3^ PFU mL^−1^ (30 min) and ~10^2^ PFU mL^−1^ (60 min) of these two phages. Regarding the other viruses (Shi22, Shi30, Shi33, Shi34, Shi40, Shi88, Shi93, and Shi113), ~10^2^ PFU mL^−1^ particles survived after a 30 min incubation period, while after 60 min all of them were completely inactivated. In general, the higher the concentration of phages the more significant the reductions in target bacteria [23]. Furthermore, phages at MOIs of 1, 10, and 10^2^ significantly reduced *S. Typhimurium* on lettuce, with the reductions increasing as the phage concentration increased [24]. In addition, when the viability was compared between the treatments with the different organic acids evaluated, results indicated that phages were highly resistant to acetic acid (pKa = 4.7) and equally resistant to lactic (pKa = 3.8) and citric (pKa = 3.1) acids. Only AShi and Shi3 showed a differential resistance to lactic and citric acid (Figure 4).

### 3.3. Phage Viability Against Biocides

Figure 5 shows the viability of *Shigella* phages against several concentrations of sodium hypochlorite. Six (AShi, Shi3, Shi22, Shi30, Shi33, and Shi88) out of the ten phages tested showed a high resistance to this biocide since they endured at concentrations up to 100 ppm for 10 min. Furthermore, particles of AShi (10^2^ PFU mL^−1^) and Shi22 (10^1^ PFU mL^−1^) were detected at the highest concentration of hypochlorite assayed after 1 min of incubation. On the other hand, Shi34, Shi40, Shi93, and Shi113 were the most sensitive ones to this biocide. Their viabilities were significantly affected after the shorter period of incubation (1 min) at the lowest concentration (50 ppm) assessed, and almost completely inactivated (10^1^ PFU mL^−1^) after 1 min at 100 ppm.

Next, phage viability against ethanol was evaluated (Figure 6). At 10% of ethanol, phages proved to be highly resistant since no significant reduction in PFU mL^−1^ counts was observed. At a higher concentration of ethanol (70%), the most resistant phages were AShi, Shi40, Shi93, and Shi33. Namely, AShi, Shi40, and Shi93 particles were detected after 30 min, however, after 60 min of incubation these phages were completely inactivated. On the contrary, phage Shi33 showed the highest resistance against this particular biocide since a great number of particles (10^3^ PFU mL^−1^) remained viable up to 60 min. The rest of the phages tested (Shi3, Shi22, Shi30, Shi34, Shi88, and Shi113) showed a high sensitivity to this biocide and only 10^3^ PFU mL^−1^ particles were detected after 15 min. At longer incubation times, all of them were completely inactivated. Furthermore, at the highest concentration of ethanol (96%) the ten phages were completely inactivated within 15 min, as also was observed after 24 h of incubation at 10 and 70% of ethanol.

All the phages of *Shigella* showed a high resistance to QAC (Figure 7). The viability of the ten phages was essentially unaffected at 2% after a 30 min period. This was also true for Shi22 after 60 min, whilst for the other nine phages significant reductions were observed after the same period of time (60 min). At 3% of QAC, ~10^3^–10^4^ PFU mL^−1^ remained viable for most phages evaluated within 15 min. As the exposure time increased, the phage counts decreased up to 10^1^ (Shi33, Shi34, Shi88), 10^2^ (Shi40, Shi93, Shi113), and 10^3^ (AShi, Shi3, Shi22, Shi30) PFU mL^−1^, an acceptable level of survival of phages when challenged against biocides. QAC at a concentration of 4% reduced phage counts below the detection limit within 15 min, as also was observed at the last time of incubation (24 h) for all the concentrations of QAC tested.

Figure 8 shows the viability of *Shigella* phages against several concentrations of hydrogen peroxide. A similar behavior was found for most viruses evaluated when they were challenged against H_2_O_2_. At the concentration of 2%, reductions ranged from 1 to 4 log_10_ PFU, while 10^4^–10^5^ and 10^1^–10^2^ particles remained viable after 30 min and 60 min, respectively. At 3%, a higher viability reduction was observed for the ten phages, however, counts only dropped from ~10^4^ PFU mL^−1^ (15 min) to 10^1^–10^2^ PFU mL^−1^ after a 30 min exposure. After 60 min, complete phage inactivation was achieved since no viable particles could be detected. Phage Shi93 proved to have the highest sensitivity to this biocide. Only 10^3^ and 10^2^ particles were detected after 15 min against 2% and 3% of H_2_O_2_, respectively, while at longer exposure times, a complete inactivation was observed. At 4%, complete inactivation of *Shigella* phages occurred within 15 min of incubation.

## 4. Discussion

To determine if the ten *Shigella* phages that are being studied can withstand the conditions found in a food matrix and be useful as a biocontrol tool in agriculture, their viability against different preservatives was assessed. A complete inactivation was observed at low (2%) and high (4%) concentrations of lactic acid, whereas phage particles could be detected after only 15 min at 2%. Regarding acetic and citric acids, all the phages tested showed a high and a moderate resistance, respectively. However, longer incubation times produced a complete inactivation of phages. In accordance with our results, Lee and coworkers found that phage HY01, which is active against *S. flexneri*, was highly resistant to acid conditions [9], suggesting that this phage is suitable for food applications. Moreover, other authors found that phages of *Salmonella enteritidis* can increase their acid tolerance when they have been previously stressed [25], thus further studies should be carried out in order to evaluate if the resistance of the eight phages can be improved. To summarize, when we compared the three different treatments, namely acetic (Figure 1), lactic (Figure 2), and citric acid (Figure 3), the most effective acid to inactivate phages was the lactic followed in effectiveness by the citric. Therefore, the *Shigella* phages evaluated could be employed in combination with chemical preservatives as suggested against other pathogens such as *Listeria* [14] and *Escherichia coli* [26]. However, further sequencing and bioinformatics analyses of our ten phages are required prior to their application to food. Sequence determination and analysis to search for harmful genes and endotoxins are a mandatory prerequisite of applicability of phages considered for application as a biocontrol agent in food.

Next, to evaluate if our phages can withstand the action of organic acids in a food matrix, their viability was assessed in beef against the preservatives previously assayed. As is already known, *Shigella* cells can increase their acid tolerance when they have been previously exposed to acidic environments [12,27], therefore it is necessary to find phages resistant to organic acids and acidic conditions. In general, phages seem more affected by the acids in the solution than by the same acid in the matrix of the beef. This may be due to the pH values of organic acids in solution (pH ~2.20 to 2.72) being lower than the pH values achieved on the surface of beef (pH ~4.5 to 5.0) after acid application [28]. Our previous results indicated that phage viability was not significantly affected from pH 5 to pH 11, yet at pH 3 a significant reduction in viability was observed [17]. Although several biocontrol studies were conducted with phages of *Shigella* on different food matrices [14,15,20], assays in which phages were challenged against food-grade additives could not be found. Similarly, other authors found that a phage cocktail was effective at eliminating *S*. *flexneri* cells from various food [19]. However, those phages were not challenged against food additives.

Phages can be used in several applications such as the decontamination of surfaces and food processing equipment. Commonly used chemical sanitizers are capable of inactivating phages [29]. Thus, those viruses used in the food industry must be exhaustively evaluated to determine if they can withstand the conditions found in food environments. Most of the phages tested proved to be highly resistant to sodium hypochlorite, ethanol, QAC, and hydrogen peroxide since they endured at concentrations up to 100 ppm, 70%, 3%, and 3%, respectively. Although, at higher concentration of biocides and for extended periods of incubation phages were completely inactivated. Similarly, simultaneous application of bacteriophages active against *Salmonella* spp. and chemical preservatives (biocides) resulted in phage inactivation [30]. Findings presented in this work demonstrate that the use of phages and different biocides against *Shigella* strains in food applications is compatible at several biocide concentrations and exposure times. Namely, the phages of *Shigella* tested can be used in combination with 50 and 100 ppm (1 min) of sodium hypochlorite, 10 and 70% (15, 30, and 60 min) of ethanol, and 2 and 3% (15, 30, and 60 min) of QAC and H_2_O_2_. In accordance, several authors found variable resistance among other phages when treated with different biocides [31,32], although viability of *S*. *flexneri* phages had never been previously tested against the biocides evaluated in this work. Along with the determination of the genomic sequence of the phages that were suitable in the present study, it is planned to carry out assays to determine the activity of phages against the foodborne pathogens in the presence of food additives in vitro and in different food matrices.

## 5. Conclusions

The present study demonstrates that at least 10^3^ PFU of the initial number of phages survive in the presence of additives commonly applied in foods. Furthermore, the level of phages that survived was higher in the food matrix, suggesting that they can be used in combination with the additives evaluated as an additional hurdle to improve food safety. On the other hand, results indicate that these phages were highly compatible with the biocides evaluated. The longer the exposure time is, the greater the phage inactivation. However, phage AShi showed the greatest resistance against all the preservatives and biocides tested. AShi particles were detected in most of the assays performed up to the longest incubation time. To our knowledge, this is the first study in Argentina to evaluate viability of phages active against *S*. *flexneri* in food under the conditions described.

## Figures and Tables

**Figure 1 viruses-17-00474-f001:**
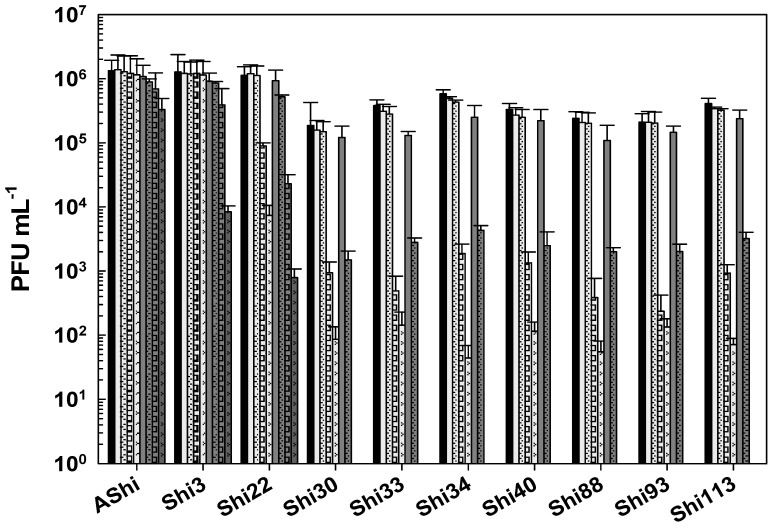
Phage viability at 25 °C without (■) and with 2% (□) and 4% (
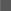
) of acetic acid after 5 min (filled), 15 min (filled and dashed), 30 min (filled and bricks), and 60 min (filled and arrows) of incubation. Values are the mean ± standard deviation (error bars) of three determinations (treatment: phage, phage concentration, incubation time) were compared against controls using Student’s *t* test at *p* < 0.05.

**Figure 2 viruses-17-00474-f002:**
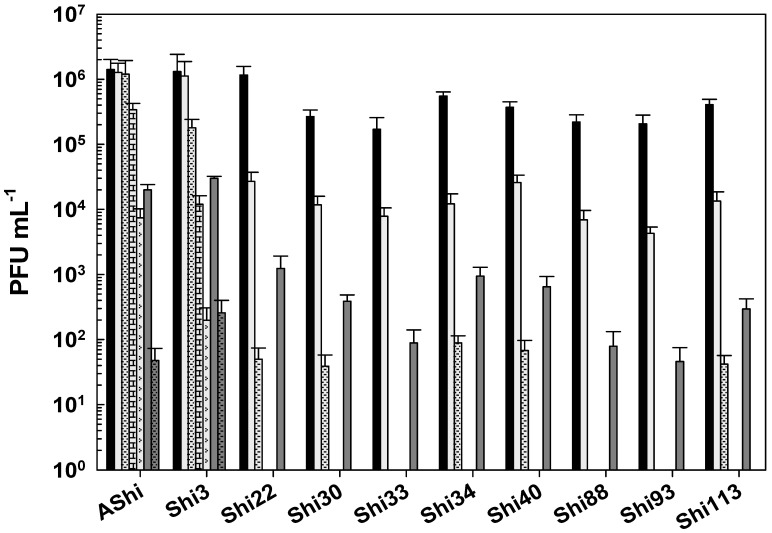
Phage viability at 25 °C without (■) and with 2% (□) and 4% (
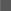
) of lactic acid after 5 min (filled), 15 min (filled and dashed), 30 min (filled and bricks), and 60 min (filled and arrows) of incubation. Values are the mean ± standard deviation (error bars) of three determinations (treatment: phage, phage concentration, incubation time) were compared against controls using Student’s *t* test at *p* < 0.05.

**Figure 3 viruses-17-00474-f003:**
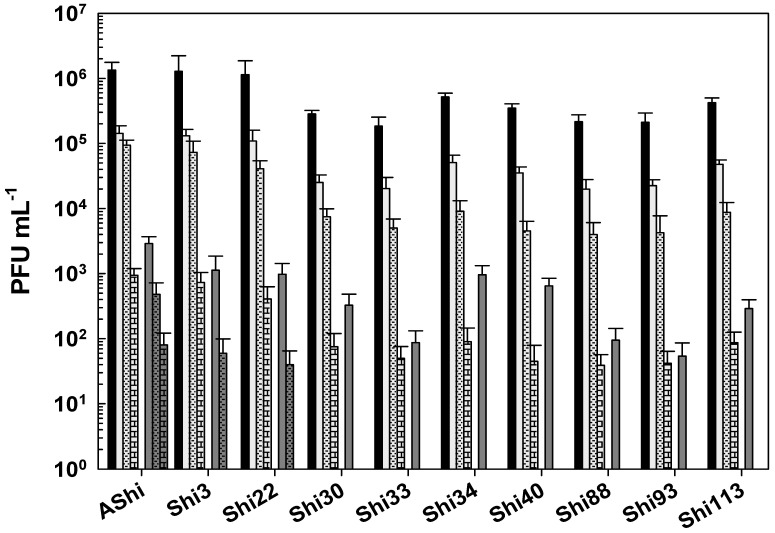
Phage viability at 25 °C without (■) and with 2% (□) and 4% (
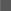
) of citric acid after 5 min (filled), 15 min (filled and dashed), and 30 min (filled and bricks) of incubation. Values are the mean ± standard deviation (error bars) of three determinations (treatment: phage, phage concentration, incubation time) were compared against controls using Student’s *t* test at *p* < 0.05.

**Figure 4 viruses-17-00474-f004:**
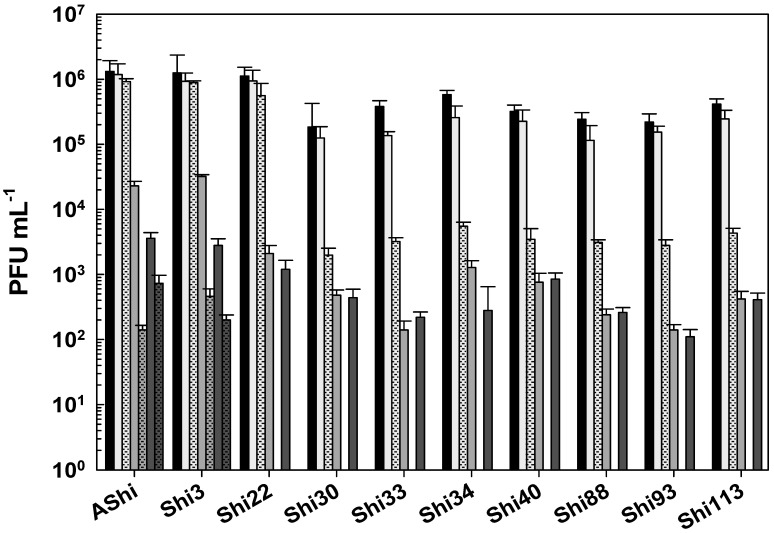
Phage viability in meat at 25 °C without (■) and with 4% of acetic (□), lactic (
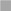
), and citric (
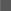
) acid after 30 min (filled), and 60 min (filled and dashed) of incubation. Values are the mean ± standard deviation (error bars) of three determinations (treatment: phage, phage concentration, incubation time) were compared against controls using Student’s *t* test at *p* < 0.05.

**Figure 5 viruses-17-00474-f005:**
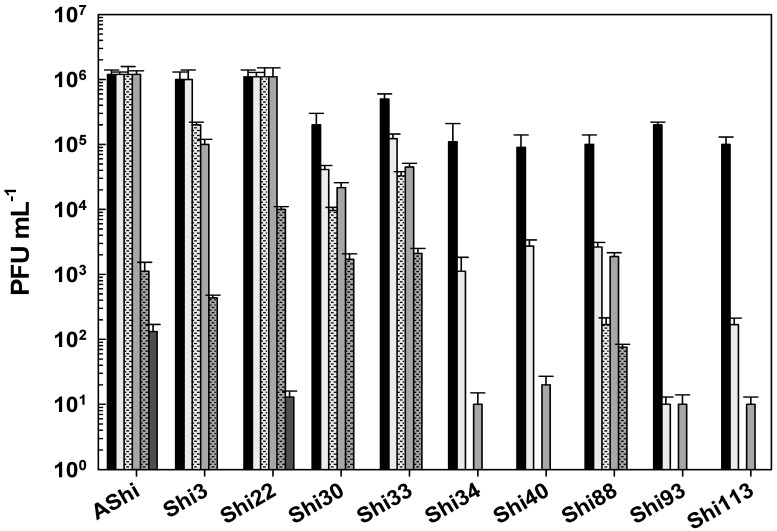
Phage viability at 25 °C without (■) and with 50 ppm (□), 100 ppm (
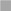
), and 500 ppm (
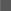
) residual-free chlorine (sodium hypochlorite) after 1 min (filled) and 10 min (filled and dashed) of incubation. Values are the mean ± standard deviation (error bars) of three determinations (treatment: phage, phage concentration, incubation time) were compared against controls using Student’s *t* test at *p* < 0.05.

**Figure 6 viruses-17-00474-f006:**
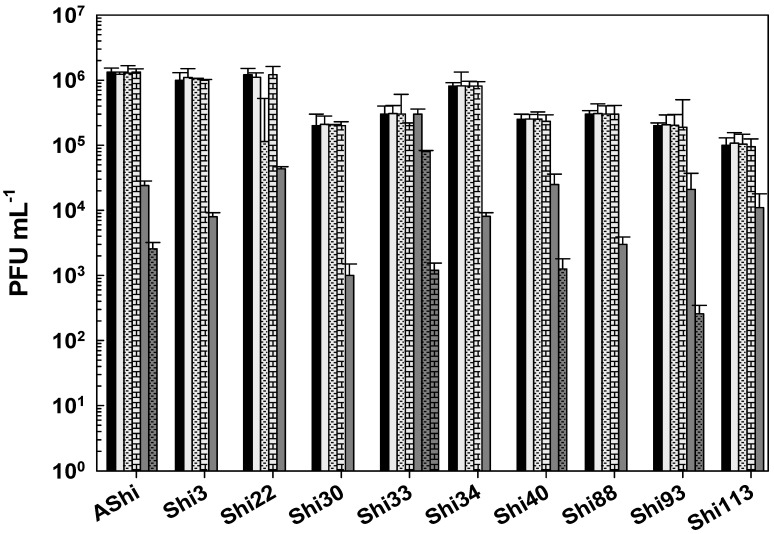
Phage viability at 25 °C without (■) and with 10% (□) and 70% (
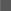
) of ethanol after 15 min (filled), 30 min (filled and dashed), and 60 min (filled and bricks) of incubation. Values are the mean ± standard deviation (error bars) of three determinations (treatment: phage, phage concentration, incubation time) were compared against controls using Student’s *t* test at *p* < 0.05.

**Figure 7 viruses-17-00474-f007:**
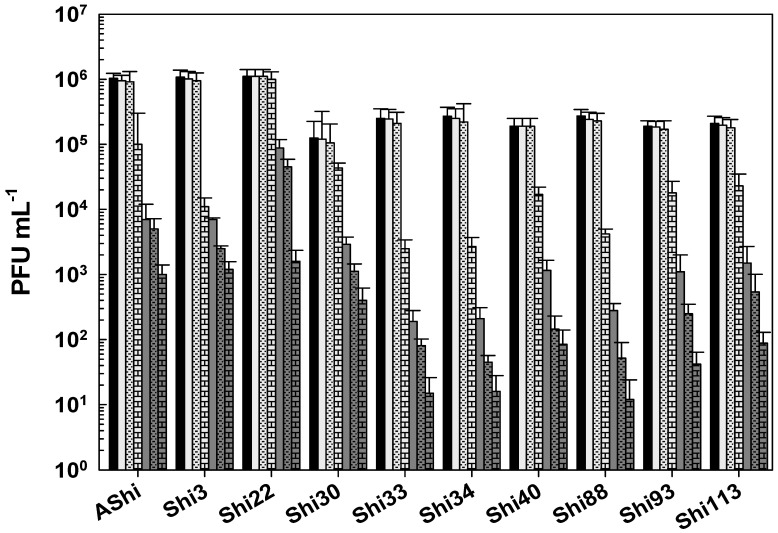
Phage viability at 25 °C without (■) and with 2% (□) and 3% (
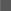
) of quaternary ammonium chloride (QAC) after 15 min (filled), 30 min (filled and dashed), and 60 min (filled and bricks) of incubation. Values are the mean ± standard deviation (error bars) of three determinations (treatment: phage, phage concentration, incubation time) were compared against controls using Student’s *t* test at *p* < 0.05.

**Figure 8 viruses-17-00474-f008:**
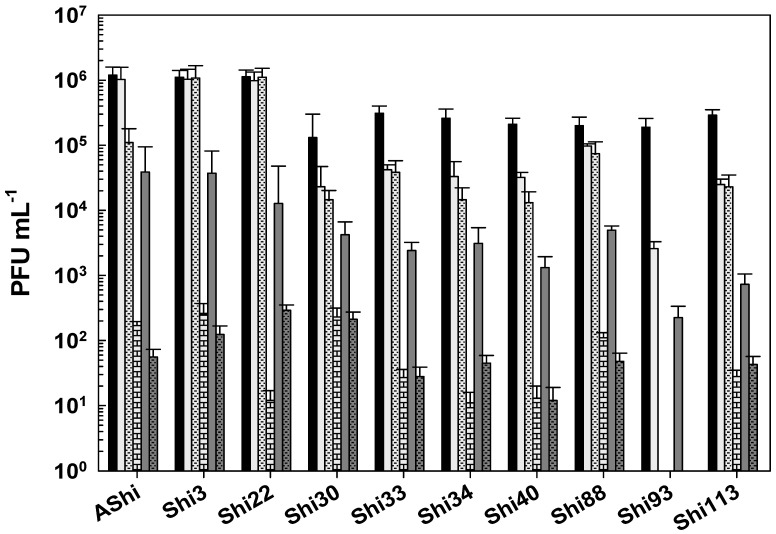
Phage viability at 25 °C without (■) and with 2% (□) and 3% (
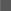
) of hydrogen peroxide (H_2_O_2_) after 15 min (filled), 30 min (filled and dashed), and 60 min (filled and bricks) of incubation. Values are the mean ± standard deviation (error bars) of three determinations (treatment: phage, phage concentration, incubation time) were compared against controls using Student’s *t* test at *p* < 0.05.

**Table 1 viruses-17-00474-t001:** Conditions tested (treatments) to evaluate phage viability.

Additive or Biocide	Concentration	Time of Incubation
acetic acid	2 and 4%	5 min, 15 min, 30 min, 60 min and 24 h
lactic acid	2 and 4%	5 min, 15 min, 30 min, 60 min
citric acid	2 and 4%	5 min, 15 min, 30 min, 60 min
acetate	2 and 4%	60 min, 120 min and 24 h
lactate	2 and 4%	60 min, 120 min and 24 h
citrate	2 and 4%	60 min, 120 min and 24 h
benzoate	0.1%	60 min, 120 min and 24 h
sorbate	0.3%	60 min, 120 min and 24 h
propionate	0.32%	60 min, 120 min and 24 h
sodium hypochlorite	50, 100 and 500 ppm	1 min and 10 min
ethanol	10, 70 and 96%	15 min, 30 min, 60 min and 24 h
QAC	2, 3 and 4%	15 min, 30 min, 60 min and 24 h
H_2_O_2_	2, 3 and 4%	15 min, 30 min, 60 min and 24 h

QAC: quaternary ammonium chloride. H_2_O_2_: hydrogen peroxide. Acetate, lactate, citrate, benzoate, and propionate (sodium salts). Sorbate (potassium salt).

## Data Availability

All data presented in this paper are available on request.

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
