# Peer review of "Evaluation of Food-Grade Additives on the Viability of Ten Shigella flexneri Phages in Food to Improve Safety in Agricultural Products"

_viruses, 2025, doi:10.3390/v17040474_

Round 1
Reviewer 1 Report (Previous Reviewer 1)
Comments and Suggestions for Authors
The authors have made a comprehensive review since the first version, especially regarding the statistics of the work.
A final observation is that the authors could include significant differences within the figures, either with letters or asterisks.
Overall, the paper seems suitable for publication.
Author Response
Reviewer #1
General Comment
The authors have made a comprehensive review since the first version, especially regarding the statistics of the work.
A final observation is that the authors could include significant differences within the figures, either with letters or asterisks.
The reviewer is right, in some cases, in graphics with less information, an asterisk can help to highlight a significant difference. However, when we added asterisks to Figure 1 (see the pdf attached), we obtained a very loaded and confusing graph. Thus, we think it is better not to include asterisks in the figures.
Overall, the paper seems suitable for publication.
We appreciate the Reviewers’ time in evaluating the manuscript, and we are thankful for their positive comments on it.

Reviewer 2 Report (New Reviewer)
Comments and Suggestions for Authors
The work of Tomat et al details the investigation of the viability of anti-Shigella phages in the presence of additives and biocides used in food industry. With the upsurge in research assessing the potential application of phages as biocontrol in foodstuff, such studies are especially warranted.
Through simple experiments, the authors demonstrated the viability of 10 previously isolated phages in the presence of a range of additives and biocides.
The manuscript is generally well-written and the figures are of good quality.
A general remark is that ’Track changes’ mode has been left on, and parts of the text remained red. The authors should remove this.
The Introduction is in need of some expansion, in order to better contextualise the study for the non-specialist readers as well. The choice of the additives investigated in the study should be justified in more detail. There should be one or more sentence and additional references on the research toward application of phages against foodborne pathogens, particularly against Shigella, with some outlook to other species as well.
There are some question and uncertainties regarding the experimental setup:
L62: What is the exact composition of ‘Tris-magnesium gelatin’? I suppose it is similar to the sodium-magnesium (SM) buffer, but neither of these is a ‘standard’ buffer, so for the sake of reproducibility and the non-specialist readers, the composition of TMG should be given at its first mention. The authors write that for the control, simply distilled water was used. I am not convinced that this is an optimal medium for the incubation of phages, and does not constitute and environmental stress in itself. It could have been simpler to use TMG for incubation of the controls without any other additive, as the authors have done in the next experiment involving foodstuff (L95).
L71: why did the authors use 25 °C as incubation temperature? Are all tested additives standardly used in foodstuff for which this is the storage temperature?
L74: is there any reference for the Argentinan food code? please include
L261: agriculture is needlessly capitalized
L276: sequence determination and analysis are indeed a prerequisite of applicability of a phage as a biocontrol agent, but please elaborate this in more detail for the sake of non-specialist readers
Another question to the authors: do they plan a follow-up study, where phages and the investigated agents are incubated together with potential target bacteria? If yes, I think it would be appropriate to mention it at the end of the Discussion section.
Author Response
Reviewer #2
The work of Tomat et al details the investigation of the viability of anti-Shigella phages in the presence of additives and biocides used in food industry. With the upsurge in research assessing the potential application of phages as biocontrol in foodstuff, such studies are especially warranted.
Through simple experiments, the authors demonstrated the viability of 10 previously isolated phages in the presence of a range of additives and biocides.
The manuscript is generally well-written and the figures are of good quality.
A general remark is that ’Track changes’ mode has been left on, and parts of the text remained red. The authors should remove this.
As suggested by this reviewer, the ’track changes’ mode has been turned off and the text in red was modified accordingly.
The Introduction is in need of some expansion, in order to better contextualise the study for the non-specialist readers as well. The choice of the additives investigated in the study should be justified in more detail. There should be one or more sentence and additional references on the research toward application of phages against foodborne pathogens, particularly against Shigella, with some outlook to other species as well.
The Introduction section was modified accordingly. The choice of the additives used has been justified as suggested. An additional sentence regarding application of phages against foodborne pathogens was added to the introduction section.
There are some question and uncertainties regarding the experimental setup:
L62: What is the exact composition of ‘Tris-magnesium gelatin’? I suppose it is similar to the sodium-magnesium (SM) buffer, but neither of these is a ‘standard’ buffer, so for the sake of reproducibility and the non-specialist readers, the composition of TMG should be given at its first mention.
TMG composition: Tris-HCl 0.05 M (pH 7.5); MgSO4 10 mM; gelatin 0.01 % w/v. The composition of TMG was added to the manuscript at its first mention as suggested.
The authors write that for the control, simply distilled water was used. I am not convinced that this is an optimal medium for the incubation of phages, and does not constitute and environmental stress in itself. It could have been simpler to use TMG for incubation of the controls without any other additive, as the authors have done in the next experiment involving foodstuff (L95).
The reviewer is right regarding the environmental stress; however, we wanted the conditions to be as similar as possible between the control and treatment, and we thought water would be best for this case. The same applies when we used TMG in food experiments.
L71: why did the authors use 25 °C as incubation temperature? Are all tested additives standardly used in foodstuff for which this is the storage temperature?
The reviewer is right since many others temperatures could be selected to evaluate phages. However, room temperature (25°C) was selected for the tests because we wanted to evaluate the viability of the phages against the additives at the time they were needed, i.e., when a food product loses the cold chain.
L74: is there any reference for the Argentinan food code? please include
The reference was included in the text and in the reference list.
L261: agriculture is needlessly capitalized
The manuscript was modified accordingly.
L276: sequence determination and analysis are indeed a prerequisite of applicability of a phage as a biocontrol agent, but please elaborate this in more detail for the sake of non-specialist readers
The manuscript was modified as suggested and a sentence was added to further explain this subject.
Another question to the authors: do they plan a follow-up study, where phages and the investigated agents are incubated together with potential target bacteria? If yes, I think it would be appropriate to mention it at the end of the Discussion section.
As suggested by this reviewer, a mention on this subject was added at the end of the Discussion section.

Reviewer 3 Report (New Reviewer)
Comments and Suggestions for Authors
Evaluation of Food-Grade Additives on the Viability of Ten Shigella flexneri Phages in Food to Improve Safety in Agricultural Products by Tomat et al. is a succinct article on using bacteriophages as biocontrol agents in agriculture to improve food safety. They carried out phage viability assays against food additives and biocides, phage viability against food additives in food. This article demonstrated that acetic acid was the most phage-friendly treatment evaluated. I therefore proposed the publication of this manuscript without reservation..
Author Response
Reviewer #3
Evaluation of Food-Grade Additives on the Viability of Ten Shigella flexneri Phages in Food to Improve Safety in Agricultural Products by Tomat et al. is a succinct article on using bacteriophages as biocontrol agents in agriculture to improve food safety. They carried out phage viability assays against food additives and biocides, phage viability against food additives in food. This article demonstrated that acetic acid was the most phage-friendly treatment evaluated. I therefore proposed the publication of this manuscript without reservation.
We appreciate the Reviewers’ time in evaluating the manuscript, and we are thankful for their positive comments on it.

Round 2
Reviewer 2 Report (New Reviewer)
Comments and Suggestions for Authors
The authors have amended the manuscript with all the suggested changes. One minor remark is that for the sake of clarity, in L43 'our country' should be changed to Argentina, but this could be done through the proofreading.
This manuscript is a resubmission of an earlier submission. The following is a list of the peer review reports and author responses from that submission.
Round 1
Reviewer 1 Report
Comments and Suggestions for Authors
General Comment
The study presents an interesting investigation into the use of phages and their interactions with additives and biocides. However, although the authors provided a subsection detailing the statistical methods employed, the results section does not clearly incorporate these methods. This lack of integration makes it difficult to determine whether the results are statistically supported or are based on subjective interpretation by the authors. Furthermore, the presentation of the results could be improved by merging the graphs, facilitating clearer and more direct comparisons between agents, phages, and exposure times.
Introduction:
Line 37: Replace with "this bacterium".
Lines 44-49: Add a reference.
Line 52: Change to: "challenge and adsorption assays"
Line 57: Change to: "Although Shigella phage studies have been conducted on various foods [16-18], those focused on evaluating the viability of S. flexneri phages challenged with a combination of additives OR everyday biocides have not been documented." You did not use additives and biocides in combination in this study.
Material and methods:
Line 63: Replace with: "were reactivated in trypticase soy broth at 37°C…"
Line 67: Merge subsections 2.2 (Phage viability against food additives) and 2.4 (Phage viability against biocides) into a single topic. Additionally, create a table summarizing all conditions tested (e.g., additives/biocides, concentrations, incubation times, and results) to improve understanding of the variables used.
Line 105: The statistical analysis used is flawed and requires full review. The authors do not specify what comparisons were made. It is unclear whether: a) Exposure time within the same organic acid was assessed using a single phage. b) Tests compared different organic acids for a single phage. or c) Performance of the same acid on all phages was compared.
Furthermore, the authors need to perform a normality test before assuming that Student's t-test is the most appropriate. There is a likely chance that the data may require a nonparametric analysis rather than a parametric one.
Results:
Line 116: The authors used the term “significant,” but when reviewing the graphs, no statistical information is presented, not even in the figure legends. As noted in the previous comment, the authors should review the entire statistical analysis of the study.
Line 124: Why did the authors choose bar graphs instead of line graphs? I suggest creating only two graphs: a) One for 2% and one for 4% organic acid concentrations. b) Combine Figures 1, 2, and 3 into these two consolidated graphs, including all results for organic acids. c) Use the line graph to demonstrate the variation in phage viability over time (5, 15, 30, and 60 minutes).
Line 143: What criteria was used to define “moderate resistance”?
Line 153: Was the efficacy of phage inactivation based on any statistical criteria? You tested 10 phages with each acid — how did you determine the order of effectiveness as lactic, citric, and acetic acid?
Line 183: This is the only figure where a visual comparison of the effects of acids on different phages is possible. However, statistical analysis is lacking.
Discussion:
Line 250: What criteria were used to define what is considered “high” and “moderate”?
Line 259: Revise the sentence to: “…and citric acid (Figure 3), the most…”
Line 264: Replace “meat” with “beef” for specificity.
Line 273-275: Is there really no research on food-grade additives with phages, even for other pathogens?
Conclusion:
Lines 297-304: The conclusion needs to be completely rewritten for clarity and to better reflect the findings of the study. For example: Highlight which phage demonstrated the greatest resistance and under what conditions (e.g., specific organic acids and exposure times). Clearly summarize the differences observed between the organic acids in terms of their effectiveness in inactivating phages. Discuss the significance of exposure time on phage viability.
Line 304: The reference to “region” is vague and needs clarification. Specify that this is the first study in Argentina.
Comments on the Quality of English LanguageNo comment.
Reviewer 2 Report
Comments and Suggestions for Authors
As biocontrol agents, bacteriophages can be applied to food preservation. The Authors addressed the possible use of food safety phages along with a number of chemical preservatives. The literature of the kind may advance the applied science of bacteriophage-based food preservation or provide the basis for regulatory approval of particular product [1]. Unfortunately, the MS in its current form does neither of the two.
The studied bacteriophages were originally published in the Ref. 14. The corresponding article does not exist. Obviously, Ref. 14 mentions in fact “Physicochemical characterization of ten newly isolated phages against the foodborne pathogen Shigella flexneri” with another title and article number (https://doi.org/10.1111/jfpp.16818). As can be seen from this original publication, no whole genome data on the studied phages is available.
Comment 1, from the scientific point of view. It is unclear if some studied phages actually belonged to the same isolate. Also, there is no way of reproducing the study results or reusing them in some form of knowledge synthesis. The bacteriophage diversity is vast. How it can be understood, which particular bacteriophage taxa possess the described properties? I strongly recommend that the whole genome sequences of bacteriophage be obtained and deposited in a publicly available database. Taxonomic identity of the bacteriophages and the life cycle strategies must be ascertained. The genomic similarity of the studied phages should be assessed.
Comment 2, from the practical point of view. At the level of law, the need for studying the whole genome sequences of food safety bacteriophages is acknowledged worldwide. In the USA, the Code of Federal Regulations implies obligatory search for a number of harmful genes [2, 3]. For example, please see a GRAS notice [4]. The Europeans additionally request taxonomic identification [5]. Therefore, potentially harmful genes should be identified, e.g. with the ABRicate tool.
Comment 3. In some experiments, the titre of Shigella flexneri phages decreased to 102-104 PFU/mL and still considered to be significant (e.g., Lines 26-27, 166-167). Phage therapy literature puts an emphasis on phage concentration, and the titre at 104 PFU/mL is generally referred to as low and inappropriate [6, 7]. Find several relevant literature sources determining the efficacy of phage preparations on food in relation to bacteriophage titre and employ this information in your residual titre assessments.
Minor
Line 25. Write the name QAC in full.
Line 60. Please correct the isolate designation to ATCC 12022 (with space) [8]
[1] Generally Accepted Scientific Knowledge in Applications for Drug and Biological Products: Nonclinical Information; Draft Guidance for Industry; Availability https://www.regulations.gov/docket/FDA-2023-D-1618
[2] Code of Federal Regulations. Title 21, Chapter I, Subchapter B, Part 172, Subpart H, § 172.785 https://www.ecfr.gov/current/title-21/chapter-I/subchapter-B/part-172/subpart-H/section-172.785
[3] Code of Federal Regulations. Title 40, Chapter I, Subchapter R, Part 725, Subpart G § 725.421 https://www.ecfr.gov/current/title-40/chapter-I/subchapter-R/part-725/subpart-G/section-725.421
[4] GRAS notification of the bacteriophage cocktail PhageGuard E™ for bio-control of E. coli 0157 on beef. https://www.fda.gov/files/food/published/GRAS-757.pdf
[5] European Food Safety Authority (EFSA). EFSA statement on the requirements for whole genome sequence analysis of microorganisms intentionally used in the food chain. EFSA J. 2024;22(8):e8912. doi: 10.2903/j.efsa.2024.8912. PMID: 39135845.
[6] Nilsson AS. Pharmacological limitations of phage therapy. Ups J Med Sci. 2019;124(4):218-227. doi: 10.1080/03009734.2019.1688433. PMID: 31724901.
[7] Fedorov E, Samokhin A, Kozlova Y, Kretien S, Sheraliev T, Morozova V, Tikunova N, Kiselev A, Pavlov V. Short-Term Outcomes of Phage-Antibiotic Combination Treatment in Adult Patients with Periprosthetic Hip Joint Infection. Viruses. 2023;15(2):499. doi: 10.3390/v15020499. PMID: 36851713.
[8] https://www.atcc.org/products/12022